# SATO: Stable Text-to-Motion Framework

## ABSTRACT

Is the Text to Motion model robust? Recent advancements in Text to Motion models primarily stem from more accurate predictions of specific actions. However, the text modality typically relies solely on pre-trained Contrastive Language-Image Pretraining (CLIP) models. Our research has uncovered a significant issue with the text-to-motion model: its predictions often exhibit inconsistent outputs, resulting in vastly different or even incorrect poses when presented with semantically similar or identical text inputs. In this paper, we undertake an analysis to elucidate the underlying causes of this instability, establishing a clear link between the unpredictability of model outputs and the erratic attention patterns of the text encoder module. Consequently, we introduce a formal framework aimed at addressing this issue, which we term the **St**able **T**ext-to-M**o**tion Framework (SATO). SATO consists of three modules, each dedicated to stable attention, stable prediction, and maintaining a balance between accuracy and robustness trade-off. We present a methodology for constructing an SATO that satisfies the stability of attention and prediction. To verify the stability of the model, we introduced a new textual synonym perturbation dataset based on HumanML3D and KIT-ML. Results show that SATO is significantly more stable against synonyms and other slight perturbations while keeping its high accuracy performance. We have presented more intuitive visualizations on the anonymous website: https://anonymous.4open.science/api/repo/project-1FC7/file/SATO.html

## CCS CONCEPTS

• **Computing methodologies** → **Computer vision**.

## KEYWORDS

Human Motion Generation, Stable Text-to-Motion Framework, Robustness

**ACM Reference Format:**
Anonymous Author(s). 2024. SATO: Stable Text-to-Motion Framework. In *Proceedings of Proceedings of the 32th ACM International Conference on Multimedia (MM '24)*. ACM, New York, NY, USA, 9 pages. https://doi.org/XXXXXXX.XXXXXXX

## 1 INTRODUCTION

The Text-to-Motion (T2M) model signifies a groundbreaking and swiftly advancing paradigm with immense potential across various domains, such as video games, the metaverse, and virtual/augmented

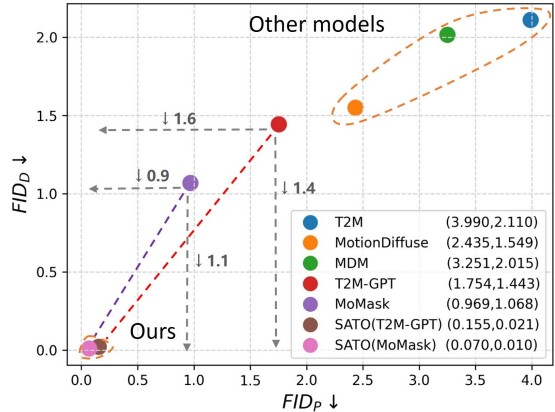

**Figure 1: Comparisons on $FID_D$ and $FID_P$. The closer the model is to the origin, the better. The arrow indicates the effect of our method on the model. Our SATO framework can make the text-to-motion model more stable.**

reality environments. This innovative approach, as evidenced by research contributions from [5, 6, 6, 8, 26, 31, 33] revolves around generating motion data directly from textual descriptions, thereby simplifying the overall process and mitigating associated time and cost overheads.

However, a fundamental challenge inherent in text-to-motion tasks stems from the variability of textual inputs [32]. Even when conveying similar or the same meanings and intentions, texts can exhibit considerable variations in vocabulary and structure due to individual user preferences or linguistic nuances. Despite the considerable advancements made in these models, we find a notable weakness: all of them demonstrate instability in prediction when encountering minor textual perturbations, such as synonym substitutions (examples and comparisons are shown in Fig. 4). This is a serious issue. The instability of the model leads to **inconsistent outputs**, with **errors in details or even entirely incorrect** motion sequence, when users input synonymous or closely related sentences. This limitation confines our model research within a narrow range of expressions, hindering the future development and practical applications of text-to-motion models. This prompts us to inquire: **What are the root causes of these issues? Are they rooted in inadequacies in textual modalities, language comprehension, or their harmonization?** Through posing these questions, elucidating this problem, and striving for a robust text-to-motion framework emerges as an urgent necessity.

Most text-to-motion models build upon pre-trained text encoders, such as CLIP [20]. Previous works have shown discrepancies in downstream tasks utilizing CLIP text encoders despite similar semantic inputs [13]. Further investigation reveals that similar phenomena occur in the text-to-motion domain. Taking the T2M-GPT [31] model as an example, several experimental findings emerge. First, we observed a close correlation between instability attention and incorrect prediction outcomes (shown in Fig. 4). Differences in attention can lead to significant disparities in text feature representations during intermediate processes. Secondly, in many instances,

 

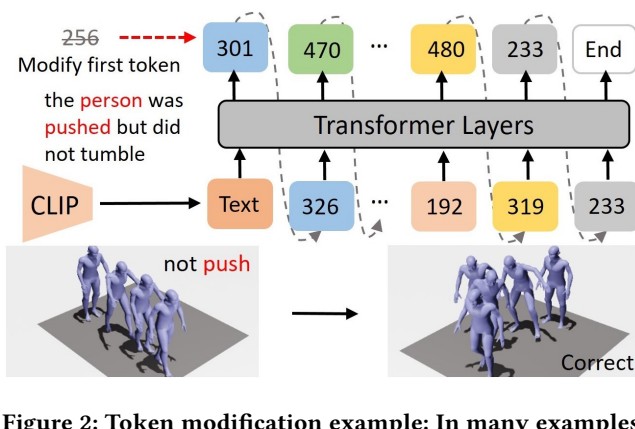

**Figure 2: Token modification example: In many examples, when the input is perturbed, the model produces an incorrect motion sequence, as shown in the bottom-left figure. When we correct the first erroneous token during the model prediction process, we obtain the correct motion sequence, as depicted in the bottom-right figure. The accuracy of the first token is crucial for the subsequent temporal predictions of the model.**

by rectifying the initial token of inaccurately predicted action sequences, subsequent accurate action sequences were obtained (see Fig. 2). Lastly, the initial motion sequence token was predicted based on the text feature. Significant differences in the text feature can lead to significant variations in the first motion sequence token. We further elucidate the aforementioned experimental findings: When perturbed text is inputted, the model exhibits unstable attention, often neglecting critical text elements necessary for accurate motion prediction. This instability further complicates the encoding of text into consistent embeddings, leading to a cascade of consecutive temporal motion generation errors. **Notably, the stability of the model manifests in the consistency of textual attention, highlighting its pivotal role in mitigating such errors.**

For a more robust text-to-motion framework, we must delve into what constitutes stability, meaning requiring us to define stability for the text-to-motion model. Intuitively, a stable attention and prediction text-to-motion model should possess the following three properties for any text input:

- Considering from a bionics perspective, it should possess a stable attention mechanism, focusing on key motion descriptions without changing with synonym perturbation.
- Its prediction distribution should exhibit stability, i.e., robustness to synonym or near-synonym substitution replacement perturbations during training and testing.
- Its prediction distribution closely resembles that of the original model in inputs without perturbation, ensuring outstanding performance.

For the first two criteria, as discussed earlier, we work on stabilizing the model's attention and predictions, both indirectly and directly, to stabilize the overall results. As for the last criterion, we emphasize the trade-off between model stability and accuracy. We aim for the model to maintain its excellent performance as much as possible. Based on these criteria, this paper presents a formal definition of a stable attention and robust prediction framework called SATO (Stable Text-to-Motion Framework).

To assess better robustness, we construct a large dataset of synonym perturbations based on two widely used datasets: KIT-ML [19] and HumanML3D [7]. It is noteworthy that even when not utilized specifically for stability tasks, our perturbed text dataset can still serve as valuable data augmentation to enhance model performance. Empirically, SATO achieves comparable performance to state-of-the-art models while demonstrating superior stability, as illustrated in Fig. 1. Extensive experimentation on these benchmark datasets, employing T2M-GPT and Momask models for verification, underscores the effectiveness of our approach. Our results reveal that we achieve optimal stability while maintaining accuracy (e.g., on T2M-GPT, HumanML3D dataset original text FID 0.157 vs **0.141**, perturbed text FID **0.155** vs. 1.754). Moreover, human evaluation results indicate a significantly reduced catastrophic error rate post-perturbation in contrast to the SOTA models, while also suggesting a subjective preference for the outputs generated by our model. In conclusion, our contributions can be summarized as follows:

- To the best of our knowledge, this is the first work to discover the instability issue in text-to-motion models. Our work formulates a formal and mathematical definition for a stable text-to-motion framework named SATO, proposes a dataset for measuring stability, and establishes relevant evaluation metrics, laying the foundation for improving the stability of text-to-motion models.
- Through extensive experimentation, we validate the effectiveness of our approach, showcasing its superiority in handling textual perturbations with comparable performance and higher stability. Additionally, we successfully strike a balance between accuracy and stability, ensuring our model maintains high precision even in the face of perturbations.
- Our work points to a novel direction for improving text-to-motion models, paving the way for the development of more robust models for real-world applications.

## 2 RELATED WORK

### 2.1 Text-conditioned human motion generation

Text-conditioned human action generation aims to generate 3D human actions based on textual descriptions. Recent mainstream work can be divided into two categories, namely VQ-VAE-based methods and diffusion models. VQ-VAE [1, 3, 4, 22, 27, 28] has achieved excellent performance in multi-modal generation tasks. ACTOR [17] proposes a Transformer-based VAE for generating motion from predefined action categories. TEMOS [18] introduces an additional text encoder based on ACTOR for generating different action sequences based on text descriptions, but mainly focusing on short sentences. Guo et al. [7] propose an autoregressive conditional VAE conditioned on the generated frame and text features, and proposed to predict actions based on the length of the text. TEACH [2] is based on TEMOS, which generates temporal motion combinations from a series of natural language descriptions and extends space for long action combinations. TM2T [8] considers not only text-to-motion tasks, but also motion-to-text tasks, and the joint training of these two tasks will be improved. T2M-GPT [31] quantizes motion clips into discrete markers and then uses a converter to generate subsequent markers. The emerging diffusion models are also changing the field of motion generation. MDM [26]

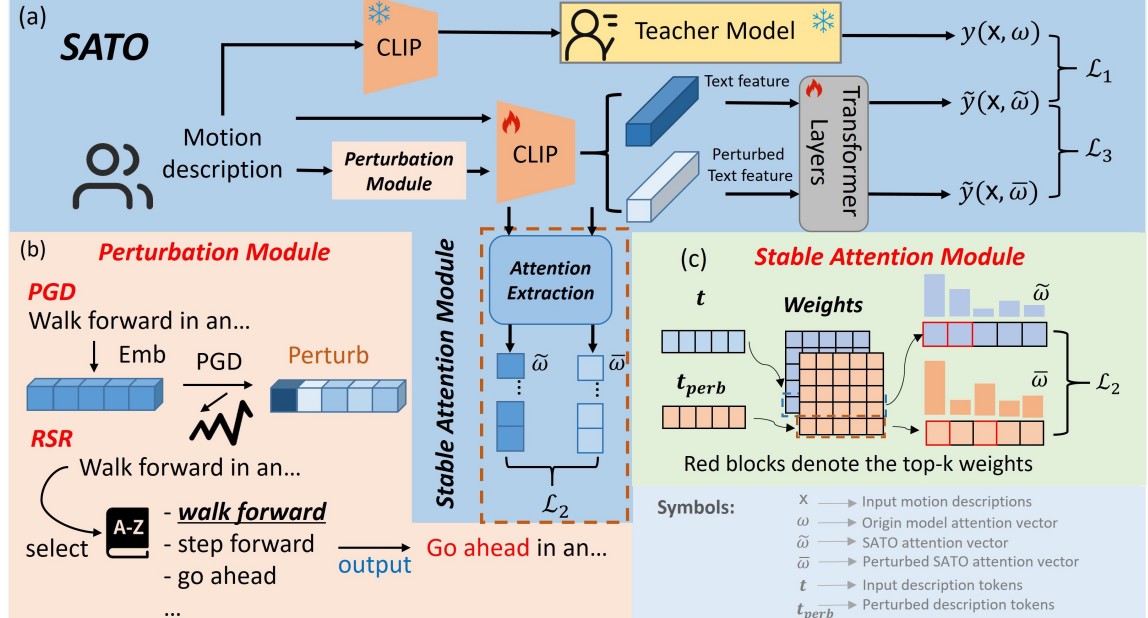

Figure 3: (a) Framework of our proposed Stable Text-to-Motion (SATO). It comprises three components: perturbation module, stable attention module, and pretrained teacher model. (b) The perturbation module encompasses two approaches for perturbation, namely Random Synonym Replacement (RSR) and Projected Gradient Descent (PGD). This module is utilized to emulate various perturbations encountered during user interactions. (c) The stable attention module aligns the top-k attention index weights before and after perturbation to stabilize the model's attention distribution. Additionally, we incorporate a frozen teacher module, solely utilized during training, to stabilize the model's motion generation capability, thus balancing the trade-off between accuracy and robustness.

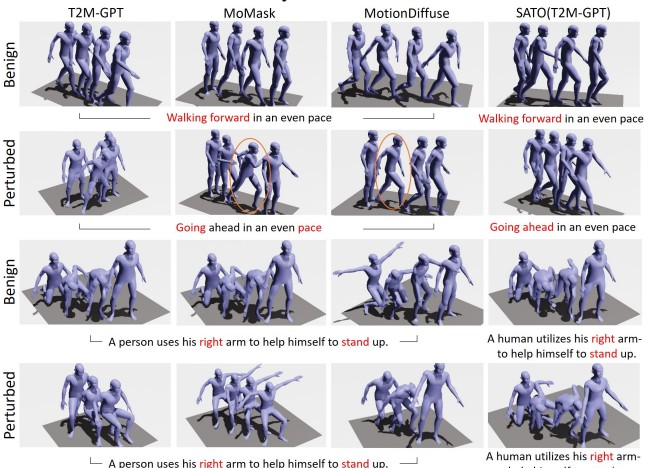

Figure 4: Visual results on user testing. SATO (T2M-GPT) refers to fine-tuning based on T2M-GPT to create SATO. Below each action sequence is the corresponding motion caption. The red color text represents the top-k attention weight words. It can be seen that the perturbation of the caption can lead to changes in the attention of the text, which can lead to catastrophic errors in the generative model. SATO has demonstrated superior stability to other models both in terms of attention and motion prediction. More visual results are provided in Supplementary Material Section 3.

uses a Transformer Encoder as the main body of prediction samples. MotionDiffuse [32] uses the DDPM architecture to generate realistic and diverse motion.

However, whether it is the diffusion-based method the VQ-VAE-based method, or even previous work such as MotionCLIP [25], the structure is based on the CLIP encoder. Although the work of TEMOS, TEACH, and Guo et al. considered the sequence length of the text and the time and space issues, they did not take into account the diversification of text raised by users. When the text is subject to slight perturbation, the model may exhibit inconsistent outputs, even leading to catastrophic errors in motion, which is a common and severe problem with these past methods. Therefore, based on these issues, this paper is the first work to consider the diversity of user-proposed texts and the first work based on the stable framework in the field of text-conditional human action generation. In this paper, we propose SATO so that the text generation results can still show strong robustness when encountering synonyms or other slight replacements or interference.

## 2.2 Stable Text-to-Motion

For the stabilization of input vector perturbations, some work has been done on stabilizing the output pattern of the model from various perspectives. Reconstructing the perturbed text with the actual input text can improve the the robustness of the text model [23], but does not guarantee the model's attention similarity before and after the perturbation. Cansu et al. [24] analyze human and machine attention to the text. However, they fail to analyze

the consistency of the descriptions before and after perturbation. Compared to the text after the perturbation, the model is more inclined to use "unfamiliar vocabulary" in human comprehension. As the "unfamiliar vocabulary" increases in a description, it interferes with the comprehension of the text. Shunsuke et al. [14] enhance the text embedding stability using adversarial learning but do not analyze the consistency between old and new attention, making it difficult to ensure that textual attention can have consistent results for different descriptive scenarios under the same semantics. Yin et al. [29] employ an adversarial robustness approach to enhance the stability of NLP models. However, these techniques are designed for ex-post interpretation of model predictions and thus cannot be applied to enhance attentional stability in the prediction phase. Different from text or visual stable attention, for the multimedia text-to-motion domain, we not only need to consider the distribution of attention weights during text embedding but also focus on the coordination performance between text and motion generation. Therefore, we need to pay attention to the consistency of textual attention before and after the description perturbation. And we also pay attention to the local importance and overall compatibility of semantic weights, to avoid the repeated generation of the emphasized part of the description, thus ignoring the coherence of the whole action.

## 3 METHOD

### 3.1 Preliminaries

**Vanilla Attention.** For text embedding, Text-to-Motion mainly uses CLIP or other text vector models [5] to encode the action description text. For the original description text, tokenization is first performed to obtain the token index vector, i.e., $\mathbf{t} \in \mathbb{R}^{n \times 1}$, where $n$ represents the number of tokens. Next, the token will be embedded by the embedding weights, i.e. $\mathbf{e} \in \mathbb{R}^{V \times d}$, where $V$ is the size of vocab, $d$ is the dimension of the embedding vectors. Therefore, when $\mathbf{t}$ goes through the embedding layer, it can obtain the corresponding embedded expression based on the token index, which is notated as $\mathbf{t_e} \in \mathbb{R}^{n \times d}$. Attention weight is to express the relationship between the query and the key, here we use Scaled Dot-Product to calculate its correlation, which is $a(\mathbf{q}, \mathbf{k}) = \text{softmax}\left(\frac{\mathbf{q}\mathbf{k}^T}{\sqrt{d}}\right) \in \mathbb{R}^{n \times m}$, where $\mathbf{k} \in \mathbb{R}^{m \times d}$ and $\mathbf{q} \in \mathbb{R}^{n \times d}$. Finally, the correlation weight is multiplied by $\mathbf{v} \in \mathbb{R}^{m \times v}$ to get the output $\mathbb{R}^{n \times v}$. Additionally, the tensors are divided into multi-heads, thus the corresponding final attention weights obtained are also the average between the individual heads, i.e. $\omega_{\mathbf{t}} = \frac{1}{h}\sum_{i=1}^{h} a(\mathbf{q}, \mathbf{k})_i \in \mathbb{R}^{n \times m}$.

**VQ-VAE Based Text-to-Motion Model.** Our objective is to produce a 3D human pose sequence $\mathbf{X} = [\mathbf{x}_1, \mathbf{x}_2, ..., \mathbf{x}_T]$, where $\mathbf{x}_t \in \mathbb{R}^d$, guided by a textual description $\mathbf{C} = [c_1, c_2, ..., c_l]$, where $T$ represents the number of frames and $d$ denotes the dimension of the motion feature. Here, $\mathbf{c}_i$ represents the $i^{th}$ word in the sentence, and $l$ is the length of the sentence. The process begins with extracting a text embedding $\mathbf{c_e}$ from input text using CLIP. Subsequently, a transformer model predicts the distribution of possible next indices $p(S_i|\mathbf{c_e}, S_{<i})$ based on the text embedding $\mathbf{c_e}$ and previous indices $S_{<i}$, where $S_i$ represents the index of the next element at position $i$ in the sequence. These predicted indices are then mapped to corresponding entries in the learned codebook, yielding latent code representations $\hat{\mathbf{z}}_i$ [27]. Finally, the decoder network decodes

these codebook entries into motion sequences $\mathbf{X_{pred}}$. The optimization objective aims to maximize the log-likelihood of the data distribution. This is achieved by denoting the likelihood of the full sequence as $p(S|\mathbf{c_e}) = \prod_{i=1}^{|S|} p(S_i|\mathbf{c_e}, S_{<i})$ and directly maximizing it: $\mathcal{L}_{\text{trans}} = \mathbb{E}_{S \sim p(S)}[-\log p(S|\mathbf{c_e})]$ [6, 31], facilitating the generation of motion sequences from input text.

**Perturbations for Texts.** To introduce effective perturbation methods for text, we consider a scenario where a perturbation $C = [c_1, c_2, ..., c_l]$ is applied to transform the original text into $C' = [c'_1, c'_2, ...c'_l]$ or perturbing the text embedding $c$ to $c'$. Several strategies have been shown to be effective in prior works, such as Greedy Coordinate Gradient (GCG) [34], Projected Gradient Descent [16] (PGD). However, due to the inherent diversity of user inputs and the presence of noise in sentences, we incorporate two distinct perturbation techniques in this study: Projected Gradient Descent (PGD) and Random Synonym Replacement (RSR). PGD finds the perturbation direction along the steepest ascent in the loss landscape, while RSR is done manually through human-designed synonym perturbations. These approaches are chosen to address the variability in user inputs and to tackle the challenges posed by noisy sentences. By employing PGD or RSR perturbations, we aim to enhance the robustness of our text-processing techniques against diverse inputs and noise.

### 3.2 Problem Formulation

**The Stability Issue in Text-to-motion Models.** The pre-trained CLIP model used as a text encoder for text-to-motion tasks has inherent limitations in maintaining stable attention for semantically similar or identical sentences, while minor perturbations are inevitable during user input. Furthermore, text-to-motion models generally lack the stable capability to handle perturbed text embeddings, leading to inconsistent predictions despite similar or identical semantic inputs. This instability renders it unsuitable for real-world applications where robustness and reliability are crucial. Addressing these issues requires us to analyze them from different perspectives.

**Attention Stability.** We first present the definition of the top-k overlap ratio for two vectors [10]. For vector $\mathbf{x} \in \mathbb{R}^n$, we define the set of top-$k$ components $T_k(\cdot)$ as:

$$T_k(\mathbf{x}) = \{i : i \in [d] \text{ and } |\{\mathbf{x}_j \geq \mathbf{x}_i : j \in [n]\}| \leq k\}.$$

For two vectors $\mathbf{x}, \mathbf{x}'$, their top-$k$ overlap ratio $V_k(\mathbf{x}, \mathbf{x}')$ is denoted as:

$$V_k(\mathbf{x}, \mathbf{x}') = \frac{1}{k \cdot |T_k(\mathbf{x}) \cap T_k(\mathbf{x}')|}. \tag{1}$$

For the original text input, we can easily observe the model's attention vector for the text. This attention vector reflects the model's attentional ranking of the text, indicating the importance of each word to the text encoder's prediction. We hope a stable attention vector maintains a consistent ranking even after perturbations. For a piece of text, demanding all attention magnitudes to be similar is overly strict. For instance, in "Walking forward in an even pace", the words "Walking", "forward", and "even" should have the most significant impact on the motion sequence. Therefore, we relax the requirement and only demand that the top-k indices remain unchanged.

**Prediction Robustness.** Even with stable attention, we still cannot achieve stable results due to the change in text embeddings when facing perturbations, even with similar attention vectors. This requires us to impose further restrictions on the model's predictions. Specifically, in the face of perturbations, the model's prediction should remain consistent with the original distribution, meaning the model's output should be robust to perturbations.

**Balancing Accuracy and Robustness.** Accuracy and robustness are naturally in a trade-off relationship [21, 30]. Our objective is to bolster stability while minimizing the decline in model accuracy, thereby mitigating catastrophic errors arising from input perturbations. Consequently, we require a mechanism to uphold the model's performance concerning the original input.

Let $y(\mathbf{x})$ denote the prediction of the original text-to-motion model, and $\omega$ denote the attention vector. Based on the discussion above, we introduce the Stable Text-to-Motion Framework (SATO) with modified prediction $\tilde{y}$ and attention vector $\tilde{\omega}$ as follows:

(1) **(Prediction Robustness)** $D_1(\tilde{y}(\mathbf{x}, \tilde{\omega}), \tilde{y}(x, \tilde{\omega} + \boldsymbol{\rho_1})) \leq \gamma_1$, for some $\|\boldsymbol{\rho_1}\| \leq R_1, \gamma_1 \geq 0$ .

(2) **(Closeness of Prediction)** $D_2(\tilde{y}(\mathbf{x}, \tilde{\omega}), y(\mathbf{x}, \omega)) \leq \gamma_2$, for some $\gamma_2 \geq 0$.

(3) **(Top-$k$ Attention Robustness)** $V_k(\tilde{\omega}, \tilde{\omega} + \boldsymbol{\rho_2}) \geq \beta$ ,for some $1 \geq \beta \geq 0, \|\boldsymbol{\rho_2}\| \leq R_2$;

Specifically, $\boldsymbol{\rho_1}$ and $\boldsymbol{\rho_2}$ represent perturbations; $R_1$ and $R_2$ is the robust radius, which measures the robust region; $D_1$ and $D_2$ are metrics of the similarity between two distributions, which could be a distance or a divergence; $\gamma_1$ measures the robustness of prediction while $\gamma_2$ measures the closeness of the two prediction distributions; $0 < \beta < 1$ is the robustness of top-$k$ indices. When $\beta$ is larger, then the attention module will be more robust; $\| \cdot \|$ is $\mathcal{L}1$ or $\mathcal{L}2$ norm.

It is worth noting that the roles of Prediction Robustness and Top-$k$ Robustness are not redundant. For instance, consider the vectors $\mathbf{v}_1 = (0.2, 0.1, \mathbf{0.4}, \mathbf{0.7})$ and $\mathbf{v}_2 = (0.3, 0.5, \mathbf{0.8}, \mathbf{1.0})$, which have the same top indices. However, the difference in their magnitudes can significantly affect the final prediction. The former affects the robustness of the prediction, while the latter emphasizes the stability of the attention vector. From a bionics perspective, the latter facilitates the model to more stably focus on crucial motion information.

## 3.3 Stable Text-to-Motion Framework

We have already proposed a rigorous definition of SATO. To build our framework (shown in Fig. 3), we use the representative T2M-GPT as the basis to provide a more concrete demonstration of SATO. And we have also verified the wide applicability of our method in MoMask [6]. To obtain a text encoder module with more stable attention, we unfreeze the CLIP module, which was originally frozen in most of the work [6, 31], and derive a minimum-maximum optimization problem with three conditions from the above three mathematical formulas, as shown in the following formula.

$$\min_{\tilde{\mathcal{W}}} \mathbb{E}_x [\lambda_1(D_2(\tilde{y}(x, \tilde{\omega}), y(x, \omega)) - \gamma_2) + \max_{\|\boldsymbol{\rho}\| \leq R} \lambda_2(\beta - V_k(\tilde{\omega}, \tilde{\omega} + \boldsymbol{\rho}))$$
$$+ \lambda_3(\max_{\|\boldsymbol{\rho}\| \leq R} D_1(\tilde{y}(x, \tilde{\omega}), \tilde{y}(x, \tilde{\omega} + \boldsymbol{\rho})) - \gamma_1)] \tag{2}$$

where $\lambda_1, \lambda_2, \lambda_3$ are hyperparameters, $\tilde{\mathcal{W}}$ represents the weight of the model. Here, we employ a maximum perturbation $\boldsymbol{\rho}$ that acts simultaneously on both factors. We need to point out that there are two challenges in the optimization: (1) **How to handle the non-differentiable function** $-V_k(\tilde{\omega}, \tilde{\omega} + \boldsymbol{\rho})$, and (2) **how to find $\boldsymbol{\rho}$ that maximizes the perturbation on $\omega$ within a certain range**.

**Stable Attention Module.** For the first issue, we need to seek an equivalent $\mathcal{L}_{\text{Topk}}$ to replace $-V_k(\tilde{\omega}, \tilde{\omega} + \boldsymbol{\rho})$. The previous discussion highlighted the necessity of considering the overlap of the previous k indices for the stability of our attention mechanism. This implies that solely relying on $\mathcal{L}_1$-norm or $\mathcal{L}_2$-norm is insufficient [10]. We need a method that is both differentiable and ensures attention to the top-k indices. One approach is to introduce the cross-distance of the values associated with the top-k indices for computation. Here, we introduce a loose surrogate loss:

$$\mathcal{L}_{\text{Topk}} = \frac{1}{2k}(\| \omega_{\zeta_k^\omega} - \tilde{\omega}_{\zeta_k^\omega} \| + \| \tilde{\omega}_{\zeta_k^{\tilde{\omega}}} - \omega_{\zeta_k^{\tilde{\omega}}} \|) \tag{3}$$

where $\zeta_k^\omega$ represents the top-k indices set of vector $\omega$, and $\| \cdot \|$ denotes a norm. In this paper, the $\mathcal{L}_1$-norm is used, which yields the best experimental results. This definition serves two purposes: it ensures the stability of the top-k indices of attention and cleverly resolves the non-differentiability issue. *We have $\omega = (0.1, \mathbf{0.3}, \mathbf{0.7})$ and $\tilde{\omega} = (\mathbf{0.5}, 0.1, \mathbf{0.2})$. We use the top-2 indices, denoted as $\zeta_2^\omega = [1, 2]$ and $\zeta_2^{\tilde{\omega}} = [0, 1]$. Using the $\mathcal{L}1$-norm, we obtain $\mathcal{L}_{Top2} = \frac{1}{4}(|[0.3, 0.7] - [0.1, 0.2]| + |[0.1, 0.3] - [0.5, 0.1]|) = 0.325.$*

**Perturbation Module.** Regarding the second issue, one approach is to introduce artificially generated high-quality synonym perturbation datasets, thereby obtaining the maximum perturbation for $\tilde{\omega}$. And for another approach, we interpret it as maximizing its susceptibility to attack for seeking the maximum perturbation for $\tilde{\omega}$. We transform this problem into solving the minimal max-attack within a certain range. Through the process of PGD [16] with n iterations, we search for the maximum attack $\boldsymbol{\rho}$.

$$\boldsymbol{\rho}_k = \boldsymbol{\rho}_{k-1}^* + \frac{r_k}{|\mathcal{B}_n|} \sum_{x \in \mathcal{B}_n} \nabla(D_2(y(x, \tilde{w}), y(x, \tilde{w} + \boldsymbol{\rho}_{k-1}^*) +$$
$$\mathcal{L}_{\text{Topk}}(\omega, \tilde{\omega} + \boldsymbol{\rho}_{k-1}^*)) \tag{4}$$

$$\boldsymbol{\rho}_k^* = \underset{\|\boldsymbol{\rho}\| \leq R}{\text{argmin}} \|\boldsymbol{\rho} - \boldsymbol{\rho}_k\|$$

where $|\mathcal{B}_n|$ represents the batch size, and $r_k$ is the step size. We utilize the gradient descent algorithm, leveraging the gradient operator $\nabla$, to iteratively update the parameters. By scaling the gradient with the step size $r_k$ and averaging it over the batch size $|\mathcal{B}_n|$, we calculate the perturbation at each iteration. Through $n$ iterations, we aim to find the maximum perturbation we desire.

**Pretrained Teacher Module.** After solving two challenging issues, we can easily interpret the first term of Equation (1). We employ a frozen pretrained T2M-GPT model as a teacher module. We aim to ensure consistency between SATO and the teacher model in predicting the original text. This is done to maintain the superior predictive performance of the original model while other modules enhance model stability during training, preventing the model from becoming overly stable and resulting in poor performance.

**SATO Optimization Goal.** we present our goal of SATO stable

loss optimization as follows:

$$\min_{\tilde{W}} \mathbb{E}_x [\lambda_1 \underbrace{(D_2(\tilde{y}(\mathbf{x}, \tilde{w}), y(\mathbf{x}, w)))}_{\mathcal{L}_1} + \lambda_2 \underbrace{\mathcal{L}_{\text{Topk}}(\tilde{\omega}, \bar{\omega})}_{\mathcal{L}_2} \qquad (5)$$

$$+ \lambda_3 \underbrace{(D_1(\tilde{y}(\mathbf{x}, \tilde{\omega}), \tilde{y}(\mathbf{x}, \bar{\omega})))}_{\mathcal{L}_3}]$$

Here $\bar{\omega}$ represents the attention vector after perturbation (PGD or RSR). In T2M-GPT (and likewise for other models), we incorporate the three mentioned losses as auxiliary attention stability losses into the original model transformer loss $\mathcal{L}_{trans}$ for fine-tuning. Eventually, we obtain:

$$\mathcal{L} = \mathcal{L}_{\text{trans}} + \lambda_1 \cdot \mathcal{L}_1 + \lambda_2 \cdot \mathcal{L}_2 + \lambda_3 \cdot \mathcal{L}_3 \qquad (6)$$

## 4 EXPERIMENTS

**Datasets.** We selected the mainstream text-to-motion datasets HumanML3D [7] and KIT-ML [19] for the evaluation of perturbed text generation and human pose generation. The purpose of the perturbation is to simulate the diversity of motion descriptions by different dimensions in the real scene. Therefore, we first define the perturbation of the motion descriptions as follows: (1) All substitutions should be randomized (different parts and number of sentences); (2) Ensure that the semantics of the motion description are consistent before and after the replacement, avoid the distraction of polysemous words; (3) There should be a clear perturbation before and after the description replacement.

In detail, we randomly select 20% of the data for each of the training, validation, and test sets of the HumanML3D dataset, respectively, analyze the replaceable words in them, and generalize the replaceable words based on their lexical properties, where the generalized categories are: nouns, adjectives, adverbs, and verbs. Then, We combine them with the context to construct a thesaurus of synonymous substitutions for each word in each lexical property and batch replacements by rules.

Our replacement rule is to traverse the word list for each motion description, randomly replacing words or verb phrases in the lexicon until two types of lexical properties have been replaced, resulting in a perturb motion description sentence. Here are some examples: *(1) "A man flaps his arms like a chicken while bending up and down." is replaced with: "A human flaps his arms like a chicken while stooping up and down." (2) "A person walks forward on an angle to the right." is replaced with: "A man walks ahead on an angle to the right." The examples in the lexicon are: "finally, ultimately, eventually," "clap, applaud, handclap,"* and so on.

We also performed a quantitative analysis of substitution as shown in Table 4, in the HumanML3D dataset, 99.13% of the motion descriptions were perturbed, and the frequency of perturbation (number of perturbed words compared to the total number of words on that description) per description amounted to 25.08%. Similarly, 97.74% of the descriptions in the KIT-ML dataset were perturbed, and the average perturbation rate reached 31.73%, which shows that the perturbation level of this perturbation strategy is fully reflected in both datasets. While keeping high perturbation, the average cosine similarity of descriptions before and after perturbation reaches 94.57% in the HumanML3D dataset, and 93.97% in

the KIT-ML dataset, which indicates that the semantics before and after perturbation have strong consistency.

**Evaluation Metrics.** In addition to the commonly utilized metrics such as Frechet Inception Distance (FID), R-Precision, Multimodal Distance (MM-Dist), and Diversity, which are employed by T2M-GPT [31], we have introduced two additional metrics based on Frechet Inception Distance to further assess the stability of the model. Additionally, we utilize Jensen-Shannon Divergence to evaluate the stability of the model's attention. Furthermore, human evaluation is employed to obtain accuracy and human preference results for the outputs generated by the model.

- **Frechet Inception Distance [9] (FID):** We can evaluate the overall motion quality by measuring the distributional difference between the high-level features of the motions.
- **Human Evaluation:** We conducted evaluations of each model's generated results in the form of a Google Form. We collected user ratings on motion prediction, which encompassed both the quality and correctness of the generated motions. Additionally, we analyzed user preferences for pose prediction. Further details will be discussed in section 4.2.
- **Jensen-Shannon Divergence [11] (JSD):** We use JSD (Jensen-Shannon Divergence) to calculate the difference in attention vectors before and after perturbation to assess the stability of attention.

Notely, to measure the stability of the model, we will use three different *FID* input calculation methods: **(1)** *FID*: the distribution distance between the motion generated from the original text and real motion. **(2)** $FID_P$: the distribution distance between the motion generated from text after paraphrasing and real motion. **(3)** $FID_D$: the distribution distance between the motion generated from the original text and the motion generated from the text after paraphrasing. Moreover, we also employ human evaluation for cross-dataset evaluation to further analyze the performance and stability of the model. For assessing the stability of model attention, we propose using JSD. A smaller JSD value indicates greater stability of attention under perturbation. More details about the evaluation metrics are provided in Supplementary Material Section 2.

## 4.1 Experimental Setup

We adopt nearly identical settings for model architecture parameters as T2M-GPT or MoMask. Additionally, We set the batch size to 64 and utilize the AdamW [15] optimizer with hyperparameters $[\beta_1, \beta_2] = [0.9, 0.99]$. The total iteration is set to 100000 and the learning rate is 1e-4, employing a linear warm-up schedule for training all models. We respectively set $\lambda_1, \lambda_2, \lambda_3$ to 0.1, 0.2, and 0.05. For perturbation, we set $r_k$ to 0.01, the PGD step as 10, and $R$ to 0.05 when we use text embedding perturbation. Training can be conducted on a single RTX4090-24G GPU. It is worth mentioning that our method is based on fine-tuning the original model to make it more stable, without incurring any additional computation cost during the inference process.

## 4.2 Comparisons with SOTA

As with previous experiments, each experiment was repeated twenty times, and we report the mean with a 95% statistical confidence interval. Tables 1 and 2 respectively present the results of models

| Dataset | Methods | Venue | $FID\downarrow$ | $FID_P\downarrow$ | $FID\downarrow$ | R-Precision | | | MM-Dist↓ | Diversity↑ |
|---|---|---|---|---|---|---|---|---|---|---|
| | | | | | | Top1↑ | Top2↑ | Top3↑ | | |
| HumanML3D | TM2T [8] | ECCV2022 | $1.501^{\pm.017}$ | $3.909^{\pm.039}$ | $1.418^{\pm.035}$ | $0.424^{\pm.003}$ | $0.618^{\pm.003}$ | $0.729^{\pm.002}$ | $3.467^{\pm.011}$ | $8.589^{\pm.076}$ |
| | T2M [7] | CVPR2022 | $1.087^{\pm.021}$ | $3.990^{\pm.064}$ | $2.110^{\pm.039}$ | $0.455^{\pm.003}$ | $0.636^{\pm.003}$ | $0.736^{\pm.002}$ | $3.347^{\pm.008}$ | $9.175^{\pm.083}$ |
| | MotionDiffuse [32] | arXiv2022 | $0.630^{\pm.001}$ | $2.435^{\pm.067}$ | $1.549^{\pm.032}$ | $0.491^{\pm.001}$ | $0.681^{\pm.001}$ | $0.782^{\pm.001}$ | $3.113^{\pm.001}$ | $9.410^{\pm.049}$ |
| | MDM [26] | ICLR2023 | $0.544^{\pm.044}$ | $3.251^{\pm.071}$ | $2.015^{\pm.027}$ | – | – | $0.611^{\pm.007}$ | $5.566^{\pm.027}$ | $9.559^{\pm.086}$ |
| | T2M-GPT [31] | CVPR2023 | $0.141^{\pm.005}$ | $1.754^{\pm.004}$ | $1.443^{\pm.004}$ | $0.492^{\pm.003}$ | $0.679^{\pm.002}$ | $0.775^{\pm.002}$ | $3.121^{\pm.009}$ | $9.722^{\pm.082}$ |
| | MoMask [6] | CVPR2024 | $0.045^{\pm.002}$ | $0.969^{\pm.030}$ | $1.068^{\pm.029}$ | $0.521^{\pm.002}$ | $0.713^{\pm.002}$ | $0.807^{\pm.002}$ | $2.962^{\pm.008}$ | $9.962^{\pm.008}$ |
| | SATO (T2M-GPT) | – | $0.157^{\pm.006}$ | $0.155^{\pm.007}$ | $0.021^{\pm.006}$ | $0.454^{\pm.003}$ | $0.637^{\pm.003}$ | $0.738^{\pm.003}$ | $3.338^{\pm.013}$ | $9.651^{\pm.050}$ |
| | SATO(MoMask) | – | $0.065^{\pm.003}$ | $0.070^{\pm.002}$ | $0.010^{\pm.001}$ | $0.501^{\pm.002}$ | $0.697^{\pm.003}$ | $0.801^{\pm.003}$ | $3.024^{\pm.010}$ | $9.599^{\pm.075}$ |

Table 1: Quantitative evaluation on the HumanML3D. ± indicates a 95% confidence interval. SATO(T2M-GPT) refers to fine-tuning based on T2M-GPT to create SATO, and similarly, SATO(MoMask) refers to fine-tuning based on MoMask to create SATO. Red indicates the best result, while blue refers to the second best.

| Dataset | Methods | Venue | $FID\downarrow$ | $FID_P\downarrow$ | $FID_D\downarrow$ | R-Precision | | | MM-Dist↓ | Diversity↑ |
|---|---|---|---|---|---|---|---|---|---|---|
| | | | | | | Top1↑ | Top2↑ | Top3↑ | | |
| KIT-ML | TM2T | ECCV2022 | $3.599^{\pm.051}$ | $10.619^{\pm.156}$ | $4.008^{\pm.228}$ | $0.280^{\pm.006}$ | $0.463^{\pm.007}$ | $0.587^{\pm.005}$ | $4.591^{\pm.028}$ | $9.473^{\pm.145}$ |
| | T2M | CVPR2022 | $3.022^{\pm.107}$ | $8.832^{\pm.153}$ | $3.864^{\pm.119}$ | $0.361^{\pm.006}$ | $0.559^{\pm.007}$ | $0.681^{\pm.007}$ | $3.488^{\pm.028}$ | $10.720^{\pm.145}$ |
| | MotionDiffuse | arXiv2022 | $1.954^{\pm.062}$ | $5.737^{\pm.172}$ | $2.496^{\pm.106}$ | $0.417^{\pm.004}$ | $0.621^{\pm.004}$ | $0.739^{\pm.004}$ | $2.958^{\pm.005}$ | $11.100^{\pm.143}$ |
| | MDM | ICLR2023 | $0.497^{\pm.021}$ | $3.564^{\pm.894}$ | $2.331^{\pm.032}$ | – | – | $0.396^{\pm.004}$ | $9.191^{\pm.022}$ | $10.847^{\pm.109}$ |
| | T2M-GPT | CVPR2023 | $0.514^{\pm.029}$ | $2.756^{\pm.023}$ | $2.894^{\pm.016}$ | $0.416^{\pm.006}$ | $0.627^{\pm.006}$ | $0.745^{\pm.006}$ | $3.007^{\pm.023}$ | $10.921^{\pm.108}$ |
| | MoMask | CVPR2024 | $0.204^{\pm.011}$ | $2.570^{\pm.092}$ | $2.234^{\pm.101}$ | $0.433^{\pm.007}$ | $0.656^{\pm.005}$ | $0.781^{\pm.005}$ | $2.779^{\pm.022}$ | $2.779^{\pm.022}$ |
| | SATO (T2M-GPT) | – | $0.513^{\pm.006}$ | $0.581^{\pm.005}$ | $0.137^{\pm.002}$ | $0.410^{\pm.011}$ | $0.619^{\pm.005}$ | $0.736^{\pm.005}$ | $3.123^{\pm.034}$ | $10.889^{\pm.066}$ |
| | SATO (MoMask) | – | $0.234^{\pm.011}$ | $0.259^{\pm.010}$ | $0.056^{\pm.002}$ | $0.425^{\pm.006}$ | $0.649^{\pm.003}$ | $0.780^{\pm.002}$ | $2.801^{\pm.019}$ | $10.499^{\pm.090}$ |

Table 2: Quantitative evaluation on the KIT-ML. ± indicates a 95% confidence interval. SATO (T2M-GPT) refers to fine-tuning based on T2M-GPT to create SATO, and similarly, SATO (MoMask) refers to fine-tuning based on MoMask to create SATO. Red indicates the best result, while blue refers to the second best.

| Text | Model | Excellent (%) | Good (%) | Fair (%) | Poor (%) | Very poor (%) | Acc (%) | Preference (%) |
|---|---|---|---|---|---|---|---|---|
| Original text | T2M-GPT | 27.0 | 29.0 | 20.5 | 18.0 | 5.5 | 76.5 | 53.5 |
| | SATO (T2M-GPT) | 29.0 | 26.5 | 22.5 | 16.0 | 6.0 | **78.0** | |
| | MoMask | 35.5 | 28.0 | 24.0 | 9.5 | 3.0 | 87.5 | 51.0 |
| | SATO (MoMask) | 29.5 | 35.0 | 24.5 | 6.5 | 4.5 | **89.0** | |
| Perturbed text | T2M-GPT | 9.0 | 15.5 | 17.5 | 22.5 | 36.5 | 41.5 | 93.0 |
| | SATO (T2M-GPT) | 26.5 | 31.5 | 16.5 | 17.5 | 7.0 | **75.5** | |
| | MoMask | 11.0 | 14.5 | 24.0 | 14.5 | 35.0 | 49.5 | 91.0 |
| | SATO (MoMask) | 22.0 | 27.5 | 32.0 | 12.0 | 6.5 | **81.5** | |
| Cross Original text | T2M-GPT | 51.5 | 19.5 | 16.0 | 10.5 | 2.5 | 87.0 | 67.0 |
| | SATO (T2M-GPT) | 55.5 | 23.0 | 15.5 | 5.0 | 1.0 | **94.0** | |
| Cross Perturbed text | T2M-GPT | 20.0 | 13.5 | 16.0 | 22.0 | 28.5 | 49.5 | 92.0 |
| | SATO (T2M-GPT) | 44.5 | 16.0 | 24.5 | 6.0 | 9.0 | **85.0** | |

Table 3: Human evaluation and cross-dataset results on the original or perturbed text. "Excellent" means completely meets the semantic, with smooth and correct expression; "Good" means generally generates well with minor details; "Fair" means contains errors in details but is overall correct; "Poor" means overall incorrect; "Very poor" means motions and text cannot be matched at all. We believe that Excellent, Good, and Fair represent correctly generated postures, while the other two represent errors. Preference indicates human preference for the compared motions. The cross-dataset evaluation result is that the model is trained on the HumanML3D dataset, with text from KIT-ML used for testing.

| Dataset | Captions | Replacement Rate | | Co-Sim (%) |
|---|---|---|---|---|
| | | Caption (%) | Word (%) | |
| HumanML3D | 87384 | 99.13 | 25.08 | 94.57 |
| KIT-ML | 12706 | 97.74 | 31.73 | 93.97 |

Table 4: Dataset analysis. We analyzed the replacement rates (sentences, vocabulary). Additionally, we calculated the cosine similarity (Co-Sim) before and after replacement to ensure the validity of our substitutions.

on the HumanML3D and KIT-ML datasets. We compare our results with six state-of-the-art (SOTA) methods.

**Stability.** It is worth noting that all other models perform poorly on the perturbed dataset, with $FID_P$ significantly greater than $FID$, indicating that diverse representations of perturbations are fatal to the performance of these models. On $FID_P$, SATO(T2M-GPT) significantly reduced by 1.599 and 2.175 on HumanML3D and KIT-ML

respectively, while SATO(Momask) decreased by 0.899 and 2.311 respectively. Similarly, there was a significant increase in $FID_D$, with SATO(T2M-GPT) decreasing by 1.422 and 2.757 respectively, and SATO(MoMask) decreasing by 1.058 and 2.178 respectively. This suggests that SATO yields similar predictive results on both perturbed and original datasets, indicating stronger stability. We can also observe a significant reduction in the fluctuation of our model on the $FID$, $FID_P$, $FID_D$ metrics, which also reflects the stability of our approach in predictions. We further investigated the impact of our approach on the attention JSD metric. Our method exhibits stability in attention, as evidenced by experiments and visualizations provided in the supplementary material.

**Accuracy.** Although our model experiences a slight decrease in $FID$ and R-precision, we would like to point out that previous work has shown that a small decrease in $FID$ does not necessarily imply a

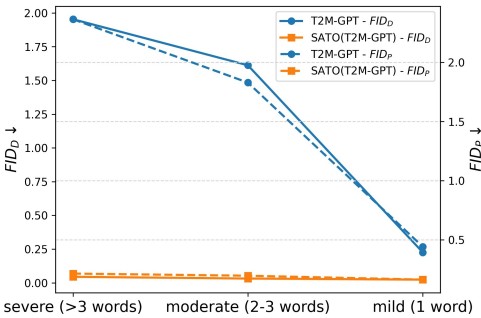

**Figure 5: Model stability evaluation under different perturbations. The x-axis represents texts with varying degrees of perturbation, while the left y-axis denotes $FID_D$ and the right y-axis represents $FID_P$. It can be observed that across all levels of perturbation, SATO (T2M-GPT) consistently outperforms T2M-GPT in terms of stability metrics. Even when subjected to significant perturbation, our model maintains excellent stability.**

decrease in generation quality [12]. Our visualization results corroborate this point, and we have additional examples from supplementary material and anonymous website to further substantiate this perspective. Furthermore, compared to the significant improvement in stability measured by the $FID$ metric, the slight decrease in $FID$ on the original text can be almost disregarded. Our visualizations and additional human evaluation experiments also demonstrate that the quality of text generated by our model on both original and perturbed text is superior to the original model. This means that our model can generate higher-quality motion sequence outputs in practical applications, with a lower likelihood of catastrophic errors occurring when presented with a broader range of textual inputs.

**Human Evaluation on the Original or Perturbed Text.** Our work's motivation is to address the catastrophic errors users encounter when using perturbed text by implementing a stable attention model. We further conduct a user study on Google Forms to validate the correctness of the model's generation. We generated 200 motions for each method using the same text pool from the HumanML3D test set as the input for all baseline models and SATO. We set up questions for users to rate the motions. Table 3 shows that SATO not only maintains or even achieves better accuracy on the original text but also ensures stability when the text is perturbed. User preferences indicate that SATO performs slightly better than the original model on the original text and significantly outperforms the original model on the perturbed text dataset. More details can be found in the supplementary material.

**Cross dataset evaluation.** To further test and evaluate the robustness and applicability of our model, we compare SATO (T2M-GPT) with T2M-GPT as examples. We conducted training on the HumanML3D dataset, using 200 original and perturbed texts from the sample kit dataset as inputs for evaluation. Table 3 illustrates that SATO (T2M-GPT) achieves higher accuracy than the baseline by 7% on original texts and by 35.5% on perturbed datasets. Evaluators also tend to favor the quality of outputs generated by SATO. Both

metrics indicate that our model demonstrates strong robustness to dataset variations. This also suggests that our approach can enhance the generalization performance of the model, enabling it to be applied in a wider range of domains.

**Overall, SATO achieves state-of-the-art stability, balancing accuracy and robustness, resolving catastrophic errors caused by synonymous perturbations.**

## 4.3 Ablation study

**Ablation Study of SATO Stability Component.** We conduct experiments on HumanML3D to evaluate the enhancements provided by our various modules in SATO,

based on T2M-GPT. In Table 5, we observe that both the Stable Attention Module($\mathcal{L}_2$) and Perturbation Module($\mathcal{L}_3$) contribute to improving stability, as evidenced by the enhancements in $FID_P$ by 1.521, 1.533 respectively, and $FID_D$ by 1.417, 1.422 respectively. The combined effect of these modules achieves optimal stability performance.

| $\mathcal{L}_1$ | $\mathcal{L}_2$ | $\mathcal{L}_3$ | FID | $FID_P$ | $FID_D$ |
|---|---|---|---|---|---|
| ✓ | ✗ | ✗ | **0.149** | 1.762 | 1.431 |
| ✗ | ✓ | ✗ | 0.187 | 0.221 | 0.026 |
| ✗ | ✗ | ✓ | 0.213 | 0.233 | 0.021 |
| ✓ | ✓ | ✗ | 0.162 | 0.173 | 0.017 |
| ✓ | ✗ | ✓ | 0.159 | 0.383 | 0.164 |
| ✗ | ✓ | ✓ | 0.198 | 0.168 | 0.012 |
| ✓ | ✓ | ✓ | 0.157 | **0.155** | **0.010** |

**Table 5: Ablation study results of SATO stability component. We conducted six separate ablation studies on three different loss functions. Bold indicates the best results.**

The inclusion of the pre-trained Teacher Module ($\mathcal{L}_1$) enhances the model's $FID$ performance, preventing excessive stability at the expense of accuracy, albeit with a potential slight decrease in stability metrics. Moreover, this module plays a crucial role by automating the selection of the best training iterations, striking a balance between robustness and accuracy, and keeping the model more evenly poised between stability and accuracy.

**Resistance to synonymous perturbation.** Based on the varying numbers of synonymous word substitutions in the test set, we categorize perturbations as mild (1 word), moderate (2-3 words), and severe (>3 words). Visualizing the results in Fig. 5, it's apparent that our model demonstrates superior stability compared to T2M-GPT across different levels of perturbation. Even when faced with severe perturbations, SATO consistently maintains excellent stability. Our model's stability metrics significantly outperform those of the original model on datasets with mild perturbations, underscoring its robustness across various degrees of perturbation. More ablation results are provided in Supplementary Material Section 4.

## 5 CONCLUSION

We identified instability issues in the text-to-motion task and introduced a novel framework to address them. In the process of building SATO, we tackled two key challenges. We also proposed evaluation metrics for this task and constructed a dataset of 55k perturbed text pairs. Our experiments demonstrate that SATO is an attention-stable and prediction-robust framework, exhibiting broad applicability across various baselines and datasets. We aim to encourage more researchers to delve into this issue, further enhancing the performance and stability of text-to-motion systems.

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
