# OpenReview forum: "SATO: Stable Text-to-Motion Framework"
_acmmm.org/ACMMM/2024/Conference — MM2024 Poster_

### Official Review · Reviewer_tLzW · 2024-04-28

**Rating:** 4
**Confidence:** 4

**Summary:**

This study identified instability issues in the text-to-motion task, wherein providing semantically similar text inputs leads to significant motion deviations. The paper introduces SATO to address this problem, consisting of three modules: stable attention, stable prediction, and a balance between maintaining accuracy and robustness. Additionally, a text synonym perturbation dataset based on HumanML3D and KIT-ML is introduced to validate the model's stability.

**Strengths:**

1. This paper is well-written, and the motivation behind the proposed method is clear and reasonable.

2. The paper identifies instability issues in the text-to-motion task, which is an interesting new direction.

3. Good performance is achieved by evaluating on both the original dataset and the perturbed dataset.

4. Comparative videos are provided to validate the effectiveness of the method.

**Limitations:**

1. There is an issue with the $FID_P$ and $FID_D$ in experimental comparison. SATO's method is trained on both the original and perturbed datasets, whereas other methods measure  $FID_P$ and $FID_D$ using the models trained only on the original data. In this scenario, the poor performance is expected. The fairest comparison would be: firstly training the T2M-GPT and MoMask on a combined dataset of the original and perturbed datasets, and then testing the $FID_P$ and $FID_D$.

2. There is a lack of discussion and citation of some text-to-motion papers related to robust text representations.

     [1] Jin et al. Act As You Wish: Fine-Grained Control of Motion Diffusion Model with Hierarchical Semantic Graphs. NeurIPS 2023.

     [2] Wang et al. Fg-t2m: Fine-grained text-driven human motion generation via diffusion model. ICCV 2023.

3. There are two errors in Table 1: the third "$FID$" should be "$FID_P$" and there is a citation error for MotionDiffuse[3].

     [3] Zhang et al. MotionDiffuse: Text-driven human motion generation with diffusion model. TPAMI 2024.

**Suitability:**

3

---

### Official Review · Reviewer_3dQ3 · 2024-05-23

**Rating:** 5
**Confidence:** 2

**Summary:**

This paper introduces a formal framework named Stable Text-to-Motion Framework (SATO) which improves the stability of text-to-motion models and contributes a new dataset.

**Strengths:**

1. The writing is clear and easy to understand.
2. The experimental setup is comprehensive and sufficient.
3. The motivation behind the work is novel and well-supported.
4. The performance of the SATO model is impressive.

**Limitations:**

The captions in Figure 4 have some errors:

1. The captions in Lines 3 and 4 should be consistent in their formatting.
2. The words in red should be different between Lines 3 and 4.

**Suitability:**

3

---

### Official Review · Reviewer_pmkf · 2024-05-27

**Rating:** 4
**Confidence:** 2

**Summary:**

This paper addresses the instability issues prevalent in current text-to-motion models. Specifically, the authors highlight that existing models exhibit inconsistent outputs when provided with semantically similar text inputs. To mitigate this issue, they propose the Stable Text-to-Motion Framework (SATO). The SATO framework first introduces perturbations to text inputs and then ensures the consistency of the output attention vector and predictions. Experimental results on the HumanML3D and KIT-ML datasets demonstrate improved stability while maintaining competitive accuracy.

**Strengths:**

1. The identification of stability issues in current text-to-motion models is crucial for ensuring reliable and predictable behavior in practical applications.

2.  The results demonstrate that the proposed method yields improved performance on two widely used datasets.

**Limitations:**

1. Lack of automatic metrics for evaluating the "stability" of final predictions. The proposed stability is only evaluated with user study and the JSD of attetion vectors.

2. The paper may not fully capture the range of possible perturbations encountered in real-world applications. Text perturbations can include misspellings, abbreviations, which are not comprehensively addressed in this study.

3. The experimental analysis does not discuss certain notable behaviors. For example, there is a lack of explanation for why diversity decreases on the HumanML3D dataset while showing significant improvement on KIT-ML with the MoMask baseline.

**Suitability:**

3

---

### Meta-Review · Area_Chair_4mL4 · 2024-06-26

**Recommendation:** Accept (Poster)
**Confidence:** 4

**Metareview:**

This paper addresses instability issues in text-to-motion models, where semantically similar text inputs lead to inconsistent outputs. The authors introduce the Stable Text-to-Motion Framework (SATO), which incorporates perturbations to text inputs to enhance output consistency in terms of the attention vector and predictions. Experimental validation on the HumanML3D and KIT-ML datasets shows that SATO improves stability while maintaining competitive accuracy. The paper is well-written and provides a clear motivation for its approach, which includes a new dataset for evaluating the model's performance against text synonym perturbations. The comprehensive experiments and the inclusion of comparative videos further support the effectiveness of the SATO framework. All reviewers agreed to accept the paper.